# Testing for SARS-CoV-2 in resource-limited settings: A cost analysis study of diagnostic tests using different Ag-RDTs and RT-PCR technologies in Mozambique

Nelmo Jordão Manjate[1], Nádia Sitoe[1☯]*, Júlia Sambo[1☯], Esperança Guimarães[1], Neide Canana[2], Jorfélia Chilaúle[1], Sofia Viegas[1], Neuza Nguenha[1], Ilesh Jani[1], Giuliano Russo[3]

1 Instituto Nacional de Saúde, Marracuene, Mozambique, 2 Malaria Consortium, Maputo, Mozambique, 3 The Wolfson Institute for Population Health, Queen Mary University of London, London, The United Kingdom

☯ These authors contributed equally to this work.

* nadia.sitoe@ins.gov.mz

**Data Availability Statement:** All datasets and software used for supporting the conclusions of

## Abstract

Early diagnosis of SARS-CoV-2 is fundamental to reduce the risk of community transmission and mortality, as well as public sector expenditures. Three years after the onset of the SARS-CoV-2 pandemic, there are still gaps on what is known regarding costs and cost drivers for the major diagnostic testing strategies in low- middle-income countries (LMICs). This study aimed to estimate the cost of SARS-CoV-2 diagnosis of symptomatic suspected patients by reverse transcription polymerase chain reaction (RT-PCR) and antigen rapid diagnostic tests (Ag-RDT) in Mozambique. We conducted a retrospective cost analysis from the provider's perspective using a bottom-up, micro-costing approach, and compared the direct costs of two nasopharyngeal Ag-RDTs (Panbio and Standard Q) against the costs of three nasal Ag-RDTs (Panbio, COVIOS and LumiraDx), and RT-PCR. The study was undertaken from November 2020 to December 2021 in the country's capital city Maputo, in four healthcare facilities at primary, secondary and tertiary levels of care, and at one reference laboratory. All the resources necessary for RT-PCR and Ag-RDT tests were identified, quantified, valued, and the unit costs per test and per facility were estimated. Our findings show that the mean unit cost of SARS-CoV-2 diagnosis by nasopharyngeal Ag-RDTs was MZN 728.00 (USD 11.90, at 2020 exchange rates) for Panbio and MZN 728.00 (USD 11.90) for Standard Q. For diagnosis by nasal Ag-RDTs, Panbio was MZN 547.00 (USD 8.90), COVIOS was MZN 768.00 (USD 12.50), and LumiraDx was MZN 798.00 (USD 13.00). Medical supplies expenditures represented the main driver of the final cost (>50%), followed by personnel and overhead costs (mean 15% for each). The mean unit cost regardless of the type of Ag-RDT was MZN 714.00 (USD 11.60). Diagnosis by RT-PCR cost MZN 2,414 (USD 39.00) per test. Our sensitivity analysis suggests that focussing on reducing medical supplies costs would be the most cost-saving strategy for governments in LMICs, particularly as international prices decrease. The cost of SARS-CoV-2 diagnosis using Ag-RDTs was three times lower than RT-PCR testing. Governments in LMICs can include cost-

this article are available at Supporting Information (S1 Dataset).

**Funding:** This work was supported by the World Health Organization (WHO), through the SARS-CoV-2 Antigen detecting rapid diagnostic test implementation projects (NS), the Global Affairs Canada and the Bill & Melinda Gates Foundation #OPP1214435 (IJ). The funders had no role in study design, data collection and analyses, decision to publish, or preparation of the manuscript.

**Competing interests:** The authors have declared that no competing interests exist.

efficient Ag-RDTs in their screening strategies, or RT-PCR if international costs of such supplies decrease further in the future. Additional analyses are recommended as the costs of testing can be influenced by the sample referral system.

## 1. Introduction

Early diagnosis and treatment of infectious diseases is fundamental to reduce the risk of virus spread, mortality rates [1–3], as well as the public expenditure in pandemic-related activities. With the onset of the SARS-CoV-2 pandemic, the World Health Organization (WHO) recommended nucleic acid amplification tests (NAATs) such as reverse transcription-polymerase chain reaction (RT-PCR) for diagnosis of SARS-CoV-2 [4, 5] and called for research on point-of-care (POC) in-vitro diagnostics (IVDs) and antigen detection rapid diagnostic tests (Ag-RDTs) [5]. Within the scope of laboratory testing policies, the Africa Centre for Disease Control (CDC) launched the "Partnership to Accelerate SARS-CoV-2 Testing" which encompassed deep-rooted approaches towards sustainable and resilient laboratory systems [6].

However, RT-PCR requires a well-equipped laboratory with advanced technology and highly trained laboratory technicians [1, 5, 7–9], which in low- and middle-income countries (LMICs), such as Mozambique, is typically available only in a few national reference laboratories [5]. In addition, in cases of high demand, RT-PCR has limited testing capacity which leads to long turnaround times, rendering it ineffective [5, 7–9], since it delays isolation and treatment to contain the spread of the virus [1]. Conversely, Ag-RDTs do not require additional infrastructure or equipment and can be operated at the point of care, with results available in less than thirty minutes [5, 9–12], which is optimal for the implementation of the test-trace-isolate strategy in resources-scarce settings.

From March 2020 to May 2021, Mozambique relied only on RT-PCR to screen for SARS-CoV-2. In June 2021, the country implemented the use of Ag-RDTs for routine diagnosis of all symptomatic suspected cases of SARS-CoV-2 in public health facilities. However, the molecular testing, RT-PCR, was recommended for all Ag-RDT negative results, for symptomatic patients [7]. At the time of this study, five types of Ag-RDT tests were in use in the public sector or under evaluation in Mozambique, and four types of RT-PCR equipment to screen for SARS-CoV-2 at the national reference laboratory at Instituto Nacional de Saúde (INS), were used.

Initially, due to resource constraints, the Ministry of Health of Mozambique opted for carrying out testing only in selected healthcare facilities (HCF), based on their human resources and infrastructure capacity to handle SARS-CoV-2 screening. Such HCFs served as points of reference for a large number of patients from neighbouring municipalities. Typically, screening was performed in tents separated from the main hospital buildings, but some in few HCF COVID-19 related outpatient care were performed in buildings, with sample collection and testing in tents. Upon arrival to the facility, patients were encouraged to follow the COVID-19 safety protocols, had the first contact with a nurse who measured his/her temperature and asked the reason for the visit. If symptoms were suggestive of SARS-CoV-2 infection, the patient was sent to the tent/consultation building. The consultation was performed by a general medicine technician (GMT) (mid-level technician) or physician. Sample collection was performed by a medium laboratory technician. In some settings, the physician/GMT was the one collecting the patient's history, counselling, prescribing the medicines and interpreting the Ag-RDT result.

In other facilities, counselling was performed by a counsellor (typically an activist/community health worker), and the analysis of test results by a laboratory technician. At the central

level National Health Laboratory (INS), samples from different HCF were received by a mid-level technician, and checked for labelling. Patient data were then recorded in the National Laboratory Information System (DISA) system, the samples sterilised in dedicated booth, and the sent to the laboratory for analysis. Sample processing were carried out by a mid-level or higher-level medical technician using different testing technologies (see measurement of costs). Amongst the four HCF included in the data collection for this study, in Matola Provincial Hospital patients were allocated more contact time with the physician. In Chamanculo General Hospital and Matola Provincial Hospital, protective equipment was used intensively.

Evidence on the costs of screening and diagnosis is particularly relevant in Mozambique, where the government is considering the implementation of community-based testing using the Community Health Workers—Agentes Polivalentes Elementares (APEs) in Portuguese. If implemented, it can be argued the strategy would yield significant efficiency gains. On the other hand, it is expected that RDTs in Mozambique provided by external donors will transition to direct public expenditures for the government. The economic evidence to make large-scale screening policy decisions is therefore needed for Mozambique, as well as other comparable LMICs. Costing data for SARS-CoV-2 diagnostics in LMICs is limited. Most studies on RDTs assess the cost of diagnosing malaria and HIV, and compare testing costs to those from laboratory analysis [11–14]. In our study, we estimate the economic cost of SARS-CoV-2 diagnosis of symptomatic suspected cases in four health facilities and one national reference laboratory in Mozambique, including both recurrent and capital costs. We present the mean unit cost of diagnosing a patient by "RT-PCR" and by "Ag-RDT" to support policy decision-making.

Our results are presented separately by test type, since the use of Nasopharyngeal and Nasal in Mozambique was implemented in different periods (June-November 2021 and December 2021 to present, respectively). The overall aim of the study is to identify the basic direct costs of each type of available COVID-19 testing, and provide the needed cost information for future cost effectiveness studies on optimalcovid testing strategies in low-income countries (LICs).

## 2. Materials and methods

### 2.1 Study settings

In Mozambique, the National Health System is structured in four levels of service provision: The "primary level" which comprises health centres and health posts and include most priority health programmes. The "secondary level", which comprises district hospitals, general hospitals and rural hospitals, generally serving more than one district and constituting the first level of referral for health services. The *primary and secondary levels* provide primary health care services. Provincial hospitals (tertiary level) and central and specialized hospitals (quaternary level) offer differentiated care, provided by specialists and represent the next referral level.

Our study was undertaken in five facilities around Maputo, Mozambique's capital city area, namely: Centro de Saúde de Marracuene (CSM)–primary level, Hospital Geral de Chamanculo (HGC) and Hospital Geral de Mavalane (HGM)–both secondary level, Hospital Provincial da Matola (HPM)–tertiary level [15], and the Instituto Nacional de Saúde (INS)–headquarters for the national public health laboratory. CSM, HPM, and the INS are in Maputo province, while the remaining two are in the Greater Maputo urban area. Such locations were chosen because of the high burden of SARS-CoV-2 with Maputo City and Province (within the 11 provinces) accounting for 50% of countries cases and admissions, and because of the importance of the facilities, key references for COVID-19 and other disease management for the respective and neighbouring municipalities. Since rural healthcare facilities do not have independent

accounting system to provide overhead and other annual costing information (depend on the district level), and that testing in the private sector differs from the public sector reality, we excluded these two settings.

## 2.2 Study design

This study used a retrospective cost analysis, from the provider's perspective using a 'bottom-up micro-costing approach' [11, 16–18]. All items needed to perform a diagnosis of SARS-CoV-2 by RT-PCR or Ag-RDT, were identified through expert interview using a standardised questionnaire for all facilities involved. The questionnaire was in form of table where the clinical pathway for screening a patient, per healthcare facility, was described, the number of health professionals involved, quantity per cadre, time spent (in minutes), drugs and supplies used, and number of units were collected (S1 Table). Overhead and capital costs were also considered in the analysis. Overhead costs were obtained from 2020 financial records from the study areas, for which we accessed consolidated financial accounts (S2 Table). For the capital assets, an inventory of items used in the screening locations was undertaken. Items like furniture, medical and non-medical equipment, and buildings (where applicable), were collected (S3 Table). All cost items were identified, quantified, valued, and the unit cost per facility was estimated [19].

## 2.3 Measurement of costs

We estimated the mean cost of screening with five Ag-RDTs types:

*Nasopharyngeal*:

a. Panbio COVID-19 Ag rapid test (Abbott, Jena, Germany, Ref: 41FK10 Lot: 41ADF115A)

b. STANDARD Q COVID-19 Ag test (SD Biosensor, Suwon-si, South Korea, Ref: Q-NCOV-01G Lot: QCO3020169I).

*Nasal-only*:

a. Panbio COVID-19 Ag rapid diagnostic test device nasal (Abbott, Jena, Germany, Ref: 41FK11)

b. COVIOS Ag COVID-19 Rapid Antigen Test (Global Access Diagnostics, United Kingdom, Ref: 11811125, Lot: CA25K-121-2)

c. LumiraDx SARS-CoV-2 Ag Test (LumiraDx, London, UK, Ref.: L016000109048, Lot.: GM2000390)

and estimated the mean cost of the confirmatory laboratory test (RT-PCR) using:

i. Analyser, Roche "**COBAS 6800**" serie 1871

ii. **Abbott m2000sp** serie 275021848

iii. Thermal cyclers, ThermoFisherScientific "**QuantStudio 5**" serie 272526175, and

iv. Thermal cyclers, ThermoFisherScientific "**QuantStudio 7 Flex**", serie 278872995.

The costs of the tests for Mozambique were obtained from the Central de Medicamentos e Artigos Médicos (CMAM, Mozambique's Central Medical Stores), INS suppliers, and international suppliers through online consultation. For the latter, we added 10% of this cost to account for logistics. The cost estimated for SARS-CoV-2 diagnosis by RT-PCR and Ag-RDT are presented in both Mozambican Metical (MZN) and the United States Dollar (USD). The 2020 mean exchange rate was used at USD 1.00 = MZN 61.47. Costs were divided into Direct

Medical Costs and Direct Non-Medical Costs. The direct medical costs included: personnel cost (staff time spent diagnosing one patient/sample) and supplies cost (reagents and consumables) used to perform a single test. The direct non-medical costs included overhead and capital assets. Medium- and long-term indirect costs were not considered in this study.

**2.3.1 Direct medical costs.** *2.3.1.1 Personnel costs.* Allocation of labor time was obtained through expert opinion with key informants (i.e., interview with the physicians/nurses, and/or laboratory technicians involved in screening), according to the following assumptions:

1. Since contact time with patients was the main parameter for labor cost, to estimate the cost of time, the clinical pathways were described, the staff involved in each activity were identified and asked to estimate the amount of time spent (in minutes) delivering services directly to patients. Minimum and maximum time was asked and the average per activity was estimated.

2. The monthly salary of each worker involved in screening was obtained from the Human Resources unit of each health facility (S2 Table). A 'salary per minute' estimate was calculated, and then multiplied by the minutes spent by each worker per activity to estimate the labor cost. For that, the monthly salary was divided by (i) the number of working days, (ii) the working day was divided by the number of working hours per day, and (iii) finally, the working hour by sixty to get the salary per minute of each worker.

For instance, since the Ag-RDT takes 15 minutes to provide the result, we measured the time the staff member took to collect the patient's demographic data, draw the sample, and disclose the result. The waiting time for the result was not counted because the staff member might have been testing other patients or be involved in other activities. For activities which could be performed by any worker with adequate training, regardless his/her academic level, we sum both (superior and medium labor cost) and divided by "2" to estimate the mean labor cost of such activity.

*2.3.1.2 Supplies and RDTs.* For each activity, key informants were asked to detail the resources and the respective amounts/quantities required to screen a patient for each type of test (e.g. personal protective equipment (PPE), laboratory kits and reagents, and other supplies), as well as the number of days/times/patients for which each item could be used. The mean number of patients tested per day, was obtained by dividing the number of Ag-RDT tests performed from June 1 to December 31, 2021, by 180 days (6 months), in each health center. To obtain the mean unit cost of diagnosis by Ag-RDT for SARS-CoV-2, we added the unit costs per health facility and divided by four (the number of health facilities). Additionally, to obtain the mean cost of diagnosis by RT-PCR, we added the unit costs of all types of nucleic acid amplification tests (NAATs) performed in different technologies (COBAS 6800; Abbott m2000sp; QuantStudio 5 or QuantStudio 7 Flex), and divided by three (the number of platform types). QuantStudio 5 and 7 use the same type and quantity of reagents. The COBAS 6800 Analyser and Abbott are Automated RT-PCR while QuantStudio 5 and QuantStudio 7 are Manual RT-PCR.

**2.3.2 Direct non-medical costs.** *2.3.2.1 Overhead costs.* Overhead costs are also called "operating costs" in the literature [19]. In this study, these included water, electricity, telephone/communication, maintenance (of buildings, vehicles, and equipment), fuels and lubricants, stationery (including printing and binding), cleaning and hygiene materials. Data on these cost categories were obtained from the 2020 financial records of each health facility. Overhead costs were attributed to each patient through the direct allocation, i.e., we divided the total overhead costs per health facility by the total annual number of patients attended to (outpatients and inpatients), to get the unit overhead cost.

*2.3.2.2 Capital costs.* These costs represent an investment in assets which are used over time and are mainly acquired at the initial phase of the activity. Such assets have an economic useful life of more than one year [19–21]. In our study, these included buildings, clinical tents for SARS-CoV-2 testing, medical and non-medical equipment, and furniture. To estimated their contribution in the unit cost of diagnosis, we found the annuitized cost of each capital item by (i) combining its useful life with a discount rate of 10% (as per the Bank of Mozambique rate), then (ii) multiplying the result (which is a decimal number) by the replacement cost (market price) of such item [10, 18]; and then (iii) sum the health facility's total annuitized costs and divided by the respective number of tests performed. For the replacement cost of buildings, we multiplied the cost per square meter ($m^2$) by the total area occupied. The cost per square meter was provided by civil construction engineers of the Provincial Heath Service, in Maputo. To capture the cost of other capital items (such as patient chairs, stainless steel consultation benches, office chairs, tables and filing cabinets), we generated an inventory in all rooms/places from where patients are screened. All items, either purchased by the government or donated were included in this costing exercise.

## 2.4 Cost analysis

We first grouped the cost items by type (personnel, supplies, overhead, and capital costs) and estimated the unit costs per healthcare facility (HCF). Since the clinical pathway/standard operation procedure for screening a patient per HCF did not change irrespective the change of the type of Ag-RDT device throughout that period (June–December 2021), we kept the cost of the remaining items constant and varied the cost of Ag-RDT device, to find different screening costs per type of Ag-RDT per HCF. Secondly, we estimated the mean unit cost per type of Ag-RDT irrespective the HCF by adding the different costs of a given Ag-RDT and divide by 4 (number of HCF), and finally, added the obtained mean unit cost per type of Ag-RDT irrespective the HCF and divided by 5 (number of Ag-RDT types) to obtain the total mean unit cost of SARS-CoV-2 diagnosis regardless the level of the HCF and the type of Ag-RDT used. The mean cost of diagnosis by RT-PCR was obtained by adding the unit costs of all types of nucleic acid amplification tests (NAATs) performed in different technologies and divided by three (the number of platform types).

## 2.5 Sensitivity analysis

To assess the uncertainty around the mean cost per test, we conducted a sensitivity analysis using various scenarios to find out different costs the government could face. We developed three scenarios to reflect changes in these key cost drivers, and plotted in Excel the overall costs against the scenarios 0 of the average costs of testing found in the five settings where we collected our data [22].

We created the following three cost scenarios:

- Scenario 1 used a less expensive labour cost (nurses/medium technicians only) to carry out all the tests (with all the other cost consitions remaining constant);

- Scenario 2 where medical supplies, specifically the RDT devices, were donated and therefore set to zero (all other conditions constant);

- Scenario 3 where we used the current international market price for the seven tests, as provided by the 2023 Global Fund Procurement Reference Pricing.

**Table 1. Unit cost per test type and per health facility for diagnosis of SARS-CoV-2 by Ag-RDT and RT-PCR.**

| Type of Test | Device Name | Ag-RDT | | | | | | | | | | | | RT-PCR | |
| --- | --- | --- | --- | --- | --- | --- | --- | --- | --- | --- | --- | --- | --- | --- | --- |
| | | CSM | | HGC | | HGM | | HPM | | Mean Cost per Type of Test | | | INS | |
| | | MZN | USD | MZN | USD | MZN | USD | MZN | USD | MZN | USD | | MZN | USD |
| *Nasopharyngeal* | Panbio COVID-19 Ag rapid test | 522 | 8.5 | 818 | 13.3 | 639 | 10.4 | 934 | 15.2 | 728 | 11.9 | | 2 414 | 39.3 |
| *Nasopharyngeal* | STANDARD Q COVID-19 Ag test | 522 | 8.5 | 818 | 13.3 | 639 | 10.4 | 934 | 15.2 | 728 | 11.9 | | | |
| *Nasal* | Panbio COVID-19 Ag rapid diagnostic test device nasal | 344 | 5.6 | 633 | 10.3 | 455 | 7.4 | 756 | 12.3 | 547 | 8.9 | | | |
| *Nasal* | COVIOS Ag COVID-19 Rapid Antigen Test | 566 | 9.2 | 854 | 13.9 | 676 | 11.0 | 977 | 15.9 | 768 | 12.5 | | | |
| *Nasal* | LumiraDx SARS-CoV-2 Ag Test | 590 | 9.6 | 885 | 14.4 | 707 | 11.5 | 1 008 | 16.4 | 798 | 13.0 | | | |
| TOTAL MEAN | | | | | | | | | | 714.0 | 11.6 | | | |

## 2.6 Ethics approval

The study protocol was approved by the Mozambique's National Bioethics Committee with approval number: 719/CNBS/20.

## 3. Results

Across the five sites, the total mean unit cost of SARS-CoV-2 diagnosis regardless the level of the health facility and the type of Ag-RDT used was MZN 714 (USD $11.60). As for the type of Ag-RDT used, the mean unit cost of diagnosing by nasopharyngeal was MZN 728 (USD $11,90) for Panbio, and MZN 728 (USD $11,90) for Standard Q, while the mean unit cost of diagnosing by nasal (Panbio, COVIOS, and LumiraDx) was MZN 547 (USD $8.90), MZN 768 (USD $12.50), and MZN 798 (USD $13.00), respectively. The mean unit cost of SARS-CoV-2 diagnosis by RT-PCR was MZN 2,414 (USD $39.2) per test, being three times higher than the cost of diagnosis by Ag-RDT. Table 1 summarises the unit costs of diagnosis by Ag-RDTs and RT-PCR, and the detailed quantities of resources needed for diagnosing by each method/type of test are displayed in the S4 Table.

In both settings, health facility and laboratory, supplies were the main cost drivers followed by personnel and overhead. Table 2 provides details on the major driver for the cost difference among the different Ag-RDTs types.

Fig 1 shows the average value of cost components for each tests, regardless to health facilities. The full costs per device test and per health facility are presented in the S4 Table.

### 3.1 Personnel cost & supplies (reagents and consumable)

Amongst the four health facilities, HP Matola presented the highest labor cost, at least 2.8 times higher than others. The long contact time between the doctor and patient made the labor cost expensive at this location. Supplies amounted to more than 50% of the mean unit

**Table 2. Summary of resources per type of test and costs for Ag-RDT tests (USD 2020).**

| Type of Cost | Item | CS Marracuene | HG Chamanculo | HG Mavalane | HP Matola |
| --- | --- | --- | --- | --- | --- |
| *Direct Medical Cost* | Personnel | 1.2 | 1.1 | 1.4 | 3.9 |
| | Supplies | 6.3 | 7.8 | 6.9 | 8.6 |
| *Direct non-Medical Cost* | Overhead | 0.2 | 3.4 | 1.8 | 2.2 |
| | Capital | 0.8 | 1.0 | 0.3 | 0.5 |
| TOTAL | | 8.5 | 13.3 | 10.4 | 15.2 |

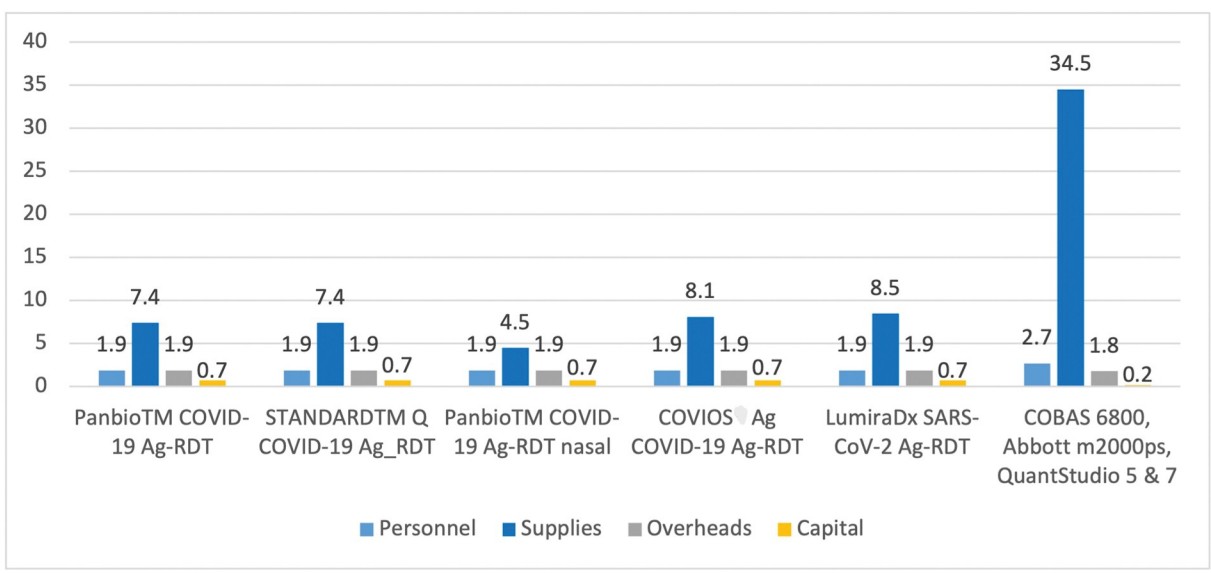

**Fig 1. Average value of cost components composing the unit cost (in USD) of SARS-CoV-2 diagnosis per per device type.**

cost of diagnosis by Ag-RDT in each health facility. At INS, this cost component was higher and amounted to 88% of the total cost.

**Overhead & capital costs.** Overhead had a relatively higher burden in HG Chamanculo and HP Matola. In these health facilities, operational costs such as electricity, water, fuel, and maintenance increased the cost. Capital costs did not show a significant contribution. This is because in CS Marracuene screening was performed in the actual facility buildings rather than in testing tents. In CS Chamanculo, consultation was performed in buildings while the collection of samples was in a tent. In the remaining 2 (HG Mavalane and HP Matola), screening for SARS-CoV-2 was performed in tents separated from the main hospitals' buildings. On the other hand, the slight contribution of buildings and overhead costs in the unit cost at the INS is due to the number of tests performed in the year of analysis (i.g., the higher the denominator, the lower the result).

## 3.2 Sensitivity analysis

We created three cost scenarios where we used a less expensive labour cost (nurses/medium technicians only) to carry out all the tests (scenario 1), another where medical supplies, specifically the RDT devices, were donated and therefore set to zero (scenario 2), and final scenario where we used the current international market price for the seven tests (scenario 3). Fig 2 shows the results of varying such costs against the original scenario 0 of the costs collected on the field.

Our analysis shows that replacing physicians with nurses or medium technician for screening does not carry a substantial reduction in the costs of administering COVID-19 tests (scenario 1). Conversely, the sensitivity analysis shows that the mean unit cost of screening is higly sensitive to price of test-kits included in the medical supplies. When we reduce the cost of such kits by setting the cost of RDT devices to"zero" (as in the case of donated RDT devices with the government sustaining no purchasing cost) the unit cost of screening dropped dramatically (scenario 2). In the last scenario, the current international prices for COVID-19 tests is employed (which decreased considerably between 2021 and 2023); this last scenario sees a considerable fall of the difference between testing with RT-PCR and the other testing options;

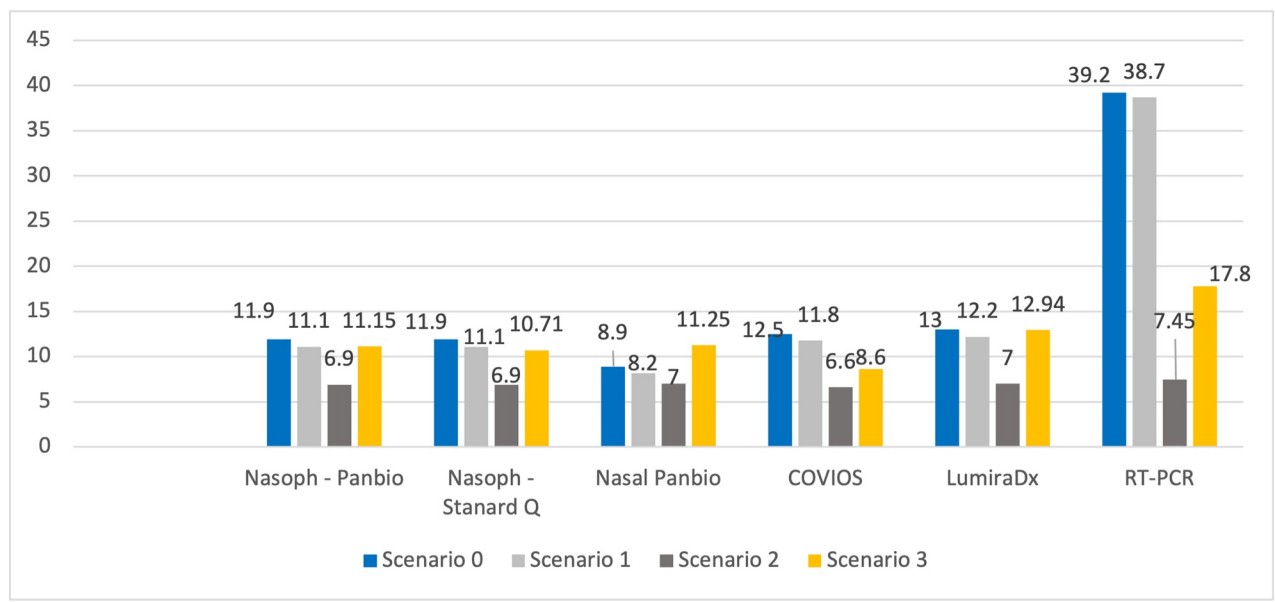

**Fig 2. Projection of testing costs under different cost scenarios for medical supplies and personnel (USD).**

testing with PCR is estimated to cost $17.8, while testing with the formerly cheaper nasal and nasopharyngeal options was costed at between $11.15 and 12.94 (Fig 2).

## 4. Discussion

This study presented the mean unit cost of SARS-CoV-2 diagnosis by Ag-RDT and RT-PCR, with the aim of providing evidence to inform decisions on implementing SARS-CoV-2 screening strategies in LMICs. In our sample of facilities in Mozambique's capital city areas, the mean unit cost of SARS-CoV-2 diagnosis by Ag-RDT was USD $11.60, while that for diagnosis by RT-PCR was USD $39.2. Supplies were the main cost driver, followed by personnel and overhead. Our study revealed that the mean unit cost of diagnosis by RT-PCR was three times higher than the cost of diagnosis by Ag-RDT, suggesting that Ag-RDT offer cost savings (USD $27.40 less per test). The sensitivity analysis on the input parameters for SARS-CoV-2 diagnosis showed that if the costs of medical supplies were set to zero–as for the case of international COVID-19 tests donations experienced by Mozambique at the start of the epidemic–the financial cost of testing with Antigens drops significantly, almost to half. This suggests the mean unit cost of testing is higly sensitive to medical supplies with RDT devices playing a crutial role, while using only mid-level technicians rather than physicians, does not reduce significantly the overall costs.

Between the antigen tests, the cost difference is small for both nasopharyngeal (Panbio and Standard Q), COVIOS, and LumiraDx. The nasal Panbio have had the smallest supply cost, it corresponded to 33.8% and 37.3% of LumiraDx and COVIOS, respectively, and 41.7% of both Nasopharyngeal. Other supply items like visors, protective glasses, and hospital hooded jumpsuits also had considerable contribution. Their use was mostly related to the health facility's rigor to fulfil with SARS-CoV-2 protocols. For example, Hospital Geral de Chamanculo (HGC) and Hospital Provincial da Matola (HPM), highly fulfilled with SARS-CoV-2 protocols, being that the latter was more rigorous in using personal protective equipment (PPE). An important personnel cost was that of HPM (USD $3.9), it was twice the cost of Centro de Saúde de Marracuene (CSM) and Hospital Geral the Mavalane (HGM), and three-fold the cost

of HGC. The difference was driven by the physician's cost since in HPM, patients spent more time with physician. Physicians performed the consultations, read the test results, and counselled the positives cases to fulfil with COVID-19 protocols and to encourage their close contacts to also fulfil with protocols and seek for testing in nearby healthcare facilities. The high cost of supplies and personnel observed in HPM might be related to the readiness of dealing with high demand and complicated cases of COVID-19.

Given the superior accuracy of PCR tests, policy makers or health systems in LICs should only consider investing more in this method to cater for false negatives/positives during pandemics, if they are benefiting from international donations of RDT devices and related supplies. This would yield great benefits as shown in scenario 2. Capital and overheads costs differences proved to be negligible, which we believe is due to the low capital costs of facilities and tents used, typical of health systems in LIC with relatively old and depreciated infrastructures. Again, from a merely economic point of view, this suggests that governments in LICs should take fully advantage of these low costs when choosing between competing testing algorithms.

We also sought to compare our results with studies around in Sub-Saharan Africa (SSA) although scarce, making it hard to put our results into context. However, we found four studies discussing on SARS-CoV-2 testing costs: the first was a modelling study involving five African countries [8], the second estimated the economic cost of implementing RDT-based testing in high risk settings in Germany [16], the third estimated the unit cost of PCR diagnosis in a national reference laboratory in Ethiopia [23], and the last estimated the clinical and economic impact of screening strategies on SARS-CoV-2 in Massachusetts [24].

The modelling study found that testing for SARS-CoV-2 by Ag-RDT and RT-PCR cost (USD $6.00) and (USD $12.00), respectively [8], differing from our study, due to the cost inputs included (costs of reagents and equipment for both types of tests) and excluded (staff costs and direct non-medical costs like overhead and buildings/tents). Our results were more consistent with those of Hurtado et al. [16] who estimated the cost of SARS-CoV-2 diagnosis by Ag-RDT including RT-PCR in two tertiary hospitals and one nursing home in Germany. They found that the mean cost of screening by Ag-RDT was €14.14 ($15.67) and €12.71 ($14.08) in the hospitals, and €14.78 ($16.38) in the nursing home, using the Mozambican 2020 mean exchange rate (dollar to Euro). Our Ag-RDT cost is close to that (USD $11.6), despite the differences in settings.

In our study, medical supplies were the main cost driver, while for Hurtado et al. were staff salaries and protective gears. In our study, personnel and supplies accounted for more than 50% of the cost of screening by Ag-RDT and 88% for RT-PCR. Our findings are similar to Hurtado et al. with regards to capital items which had a trivial contribution in both studies [16]. Our RT-PCR testing results showed to be resource consuming (USD $39.00). This is in line with a study undertaken in Ethiopia (USD $37.70) [23]. A different PCR result was found in a study undertaken in the United States (US) (USD $51.00) [24]. Two scenarios might have influenced the latter: methodology used and laboratory equipment.

Our results from the sensitivity analysis carry relevant implications for testing options in LICs. One the one hand, we show that the actual cost of testing kits are by far more important drivers than personnel and infrastructures. On the other hand, we show that if such supplies are donated or the international prices converge, there may not be such a cost difference between RT-PCR and nasal or nasal pharingean swabs. The superior accuracy of the former in detecting positive and negative cases may play a more decisive role when choosing between the available testing options.

One of the limitations of this study was related to the representativeness of our sampled settings, as we included four health facilities all from the urban areas around the capital city

Maputo. The unit cost of one of the RT-PCR reagents was not available, which may have led to an underestimation of the cost. Also, our data focuses on costs, not cost-effectiveness. Further evidence is needed for the clinical performance of different test types and swab techniques used in this analysis, in order to provide more detailed understanding on what screening strategies are more suitable for low resource settings. Finally, suspected cases with negative nasal results will still need a confirmatory laboratory test and our analysis does not incorporate the broader testing algorithm. Despite the limitations, our findings contribute to the evidence needed to define the most sustainable COVID-19 testing policies in Mozambique and in countries with similar health systems when test donations stop or new tests became available.

## 5. Conclusions

Despite wide-spread testing around the world, there is little data on costs and cost drivers for SARS-CoV-2 testing policies in low-income countries. We found that the unit cost of diagnosis by Ag-RDT in Mozambique was three times lower than by RT-PCR, strongly suggesting their use in scale from an economic point of view. In our study, supplies were the main cost drivers, followed by personnel and overhead. Capital costs had a trivial contribution to final costs. Our sensitivity analysis shown that the cost of testing was highly sensitive to supplies and that personnel cost does not differ much when replacing physicians/superior technicians by nurses/medium technicians. As international prices for available COVID-19 tests converge, their accuracy in detecting the virus will become the major determinant of testing policies in LICs.

## Supporting information

**S1 Table. Tool for clinical pathway, drugs & supplies.**
(DOCX)

**S2 Table. Tool for capital items.**
(DOCX)

**S3 Table. Tool for HR, overhead, no of patients.**
(DOCX)

**S4 Table. Statistical annex.**
(DOCX)

**S1 Dataset.**
(XLSX)

## Acknowledgments

The authors greatfully acknowledge the four health facilities managers and staff, as well as INS, for the administrative data provision and interviews.

## Author Contributions

**Conceptualization:** Nádia Sitoe, Júlia Sambo, Neuza Nguenha, Giuliano Russo.

**Data curation:** Nelmo Jordão Manjate, Giuliano Russo.

**Formal analysis:** Nelmo Jordão Manjate, Esperança Guimarães.

**Funding acquisition:** Nádia Sitoe, Ilesh Jani.

**Investigation:** Nelmo Jordão Manjate, Neide Canana.

**Methodology:** Nelmo Jordão Manjate, Giuliano Russo.

**Project administration:** Jorfélia Chilaúle.

**Supervision:** Júlia Sambo.

**Validation:** Nádia Sitoe, Júlia Sambo, Ilesh Jani, Giuliano Russo.

**Writing – original draft:** Nelmo Jordão Manjate.

**Writing – review & editing:** Nádia Sitoe, Júlia Sambo, Esperança Guimarães, Sofia Viegas, Ilesh Jani, Giuliano Russo.

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
