## [Decision Letter · Decision Letter 0]

17 Jan 2023

PGPH-D-22-01885

Testing for SARS-CoV-2 in a resource-limited setting: A comparison of costs for Antigen and RT-PCR SARS-CoV-2 diagnostic tests in Mozambique

Dear Dr. Sitoe,

Thank you for submitting your manuscript to PLOS Global Public Health. After careful consideration, we feel that it has merit but does not fully meet PLOS Global Public Health’s publication criteria as it currently stands. Therefore, we invite you to submit a revised version of the manuscript that addresses the points raised during the review process.

We look forward to receiving your revised manuscript.

Kind regards,

Sara Suliman

Academic Editor

Journal Requirements:

1. Please amend your detailed online Financial Disclosure statement. This is published with the article. It must therefore be completed in full sentences and contain the exact wording you wish to be published.

a) State the initials, alongside each funding source, of each author to receive each grant. For example: "This work was supported by the National Institutes of Health (####### to AM; ###### to CJ) and the National Science Foundation (###### to AM)."

2. Please update your online Competing Interests statement. If you have no competing interests to declare, please state: “The authors have declared that no competing interests exist.”

3. Please provide a complete Data Availability Statement in the submission form, ensuring you include all necessary access information or a reason for why you are unable to make your data freely accessible. If your research concerns only data provided within your submission, please write “All data are in the manuscript and/or supporting information files” as your Data Availability Statement.

4. Please provide separate figure files in .tif or .eps format only and ensure that all files are under our size limit of 10MB.

Additional Editor Comments (if provided):

Reviewers' comments:

Reviewer's Responses to Questions

**Comments to the Author**

1. Does this manuscript meet PLOS Global Public Health’s publication criteria? Is the manuscript technically sound, and do the data support the conclusions? The manuscript must describe methodologically and ethically rigorous research with conclusions that are appropriately drawn based on the data presented.

Reviewer #1: Yes

Reviewer #2: Yes

2. Has the statistical analysis been performed appropriately and rigorously?

Reviewer #1: I don't know

Reviewer #2: Yes

3. Have the authors made all data underlying the findings in their manuscript fully available (please refer to the Data Availability Statement at the start of the manuscript PDF file)?

Reviewer #1: Yes

Reviewer #2: Yes

4. Is the manuscript presented in an intelligible fashion and written in standard English?

Reviewer #1: Yes

Reviewer #2: Yes

5. Review Comments to the Author

Reviewer #1: Overall Recommendation

Interesting study calculating the cost of rapid antigen diagnostic testing and RT-PCR for SARS-CoV-2 in Mozambique. The major findings are that the average cost of Ag testing was ~13 USD, compared to ~39 USD for RT-PCR. I think these results are interesting and informative for policymaking, however, as it stands, I do think the analysis needs some fine-tuning, the methods clarified, and the discussion reframed prior to publication.

My major concerns (with detailed explanations below) are as follows: 1) I don’t fully understand why nasopharyngeal and nasal rapid antigen tests are divided. In practice, these procedures (and as evidenced by the costs in this study), are similar. 2) The major driver of cost seemed to be medical supplies, followed by personnel, yet there is very little commentary in the study regarding the operational system in which testing took place, which to me is critical. 3) In the methods, there are several supplies that were included that seem like they should not have been, and I also don’t see clear documentation as to how capital and overhead costs were calculated. In one section, it says that building costs were included, yet in another section it says that “screening for SARS-CoV-2 was performed in tents isolated from the main hospital buildings.” In the results, this translates to the overhead/capital costs of a national lab being similar to that of a primary health center, which I find hard to believe. 4) Figure 2 in the sensitivity analysis needs re-working, as it’s very challenging to interpret that part of the analysis. 5) clarify whether these are costs for diagnosis of asymptomatic/screening purposes, or for symptomatic SARS-CoV-2, as the costs are vastly different. 6) the discussion needs reframing to highlight the points listed below.

Detailed Recommendations

Abstract

-double or 1.5-space this for easier editing.

-“Two years after the onset…” – change this to reflect that we are now 3 years out from the start of the pandemic.

-“the reverse…” “the antigen…” – “the” is not necessary.

-“2020-21…” - Specify which dates in 2020 – 2021 the study took place, given how dynamic the RDT landscape was throughout this period.

-“four health facilities, primary, secondary and tertiary levels…” – I’m not sure what primary/tertiary levels are. Are you reference to healthcare facility levels? If yes, would clarify.

-“a RT-PCR” – the “a” is extraneous here.

-“Shown” -> “show”

-the word “average” should be changed to “mean” throughout the manuscript.

-was USD 12.90 the mean for both NP Ag-RDTS? Space-permitting, would list the specific cost for each of the two NP Ag-RDTs in the same way that you did for the nasal RDTs.

-“the average unit cost regardless the type…” – there is an “of” missing here.

-“Ag-RDT is USD 12.70” – this should be a “was”, not “is”

-“costed” -> “cost”

-“Ag-RDTs were three times” – this should say “was three times”

-“analysis” -> “analyses”

-clinical performance would influence the cost-effectiveness of the test, but not the cost, per se, so I would change this. The setting in which the test takes place could influence the cost.

-add the trademark or registered symbol to all names of tests.

Introduction

-“We found….Massachusetts.” – this entire section can be moved to the discussion.

Materials and Methods

-“Material” -> “Materials”

-please list and describe what RT-PCR test was evaluated, since there is variability in cost between these tests. I now see that this is listed later, but it would be helpful to list this early on, in conjunction with the RDTs.

-“For the later…” – should be “latter”

-“United State Dollar” – should be “States”

-Tables S1-S4: why are the Panbio and Standard Q tests grouped together in table S1, whereas the other 3 are not? I would presume that the supply cost for these two tests are different, in the same way that it is for the nasal tests? If not, this needs to be clarified.

-Where is the table similar to S1-S4 with information on the RT-PCR?

-For these S1-S4 tables, to me these seem to be results, not methods, so consider putting this in the results section.

-Supplies and RDTs – I’m not sure why the section regarding HIV and malaria testing is relevant. Consider removing. Similarly, is there a reason why malaria and HIV RDT kits are included in table S6 as a supply cost? I would remove these since you’re just evaluating the cost of covid rdts. This would imply an evaluation of a program to diagnose febrile illness with RDTs, which is not the point of this study.

-Capital Costs: somewhere, please specify whether this is diagnosis of symptomatic SARS-CoV-2, or asymptomatic screening since these protocols are different. For example, I note that in table S6, a thermometer, sphygmomanometer, stethoscope, oximeter, and weighting scale are listed as equipment. This equipment would be relevant for symptomatic patients who would require vital signs, but wouldn’t be necessary in the setting of asymptomatic community screening for example. Similarly, a lot of the capital costs may not be necessary in the setting of mobile community testing for example, where you wouldn’t need a building. This should be noted early as since it is a crucial point.

-Sensitivity analysis: “access” -> “assess”

Results

-“These findings suggest that Ag-RDT…” – this is a discussion point, not a result, and should be moved to the discussion.

-Table S5: I’m not sure what the last columns are referring to, the labor cost in mzn and usd. Is this the mean labour cost per minute, averaging that for all the workers? If yes, this needs to be clarified, because there are some individuals, like GTP’s , that are skewing this.

-Figure 1:The legend needs to be fixed – “Capital” is listed twice (blue and yellow), so I’m not sure how to interpret this graph.

-consider consolidating these results overall, specifically the way they are presented. As it stands, there are a lot of subsections with one sentence each, and it’s hard to follow.

“A sensitivity analysis modifying the two main cost drivers led to ranges from SUD 1.2 to USD 3.9….” – please clarify what this is referring to. What test? What facility? Etc.

Figure two: I’m not sure what this figure means What is minimum, average, and maximum? This all needs to be clarified.

Discussion

-as above, epidemiological context and clinical performance do not influence cost. They influence cost-effectiveness, which is beyond the scope of this study.

-overall, this discussion needs to be reframed once the issues above are addressed. I think that comparing costs in Mozambique to costs from study in nursing homes in Germany is not very informative. One the above changes are made, more interesting would be to focus this discussion on 1) what are the different costs between the different antigen tests, and between antigen testing and PCR. 2) why are costs higher in certain health settings compared to others (can link this to what the major drivers of cost are), etc. 3) this discussion would give policy makers information regarding how to choose a test, estimate costs, and modify them if needed by adjusting the testing algorithms that are deployed.

Reviewer #2: Review

Testing for SARS-CoV-2 in a resource-limited setting: A comparison of costs for Antigen and RT-PCR SARS-CoV-2 diagnostic tests in Mozambique

General comments

This is a good paper on a topical issue that may guide the choice of diagnostic tests considering the cost of various diagnostic test kits available but may benefit from further review putting into consideration the below comments.

1. The tittle does not identify the study as a cost analysis study. Kindly identify the study in the tittle

Specific comments

ABSTRACT

Under the results section, the researcher can consider presenting the costs in local currency and the dollar to attain the aim of estimating the cost for the country’s specific context at the same time allow for comparability using the dollar.

INTRODUCTION

Replace the word public contamination with spread – To narrow down the purpose of the introduction of rapid diagnosis that aims to aid in implementing the test-trace-isolate strategy which is essential for the treatment of COVID-19 cases. Reducing public contamination involved a lot of measures such as isolation which is another step after diagnosis.

The research could provide the difference between the Antigen and RT-PCR SARS-CoV-2 by even stating the advantages and disadvantages of using each. To justify the country’s implementation of Ag-RDTs for routine diagnosis of all symptomatic suspected cases as a first-line strategy before the PCR test.

Methodology

Page1: Kindly add more clarifications and the reasons why you have chosen 4 public urban facilities and ignore rural and private facilities. This may compromise the generalizability of these results

Page4: The list and details of the 5 Ag-RDTs should appear in the methodology not in the background. Remove it from introduction to avoid unnecessary repetition.

Page5: There is a confusion here between direct non-medical costs and indirect costs. Indirect costs refer to productivity loss and this can be only assessed using a societal perspective yet this study uses payer perspective. Kindly clarify this.

It would be great to include a ‘unit’ cost typology to summarize the resource use measurement of the study. For example, including intervention ‘unit’ costs, direct and ancillary service ‘unit’ costs, activity costs, and input costs in the topology.

For overhead costs, where the researcher divides the total cost of each component by the total annual number of visits (both outpatients and inpatients). The estimated cost can be difficult to generalize since for a bigger facility the formulae for calculating the overhead cost might overestimate the cost of diagnostic compared to a smaller facility with a relatively less operating cost.

The methodology is not clear. Kindly clarify the following;

a. What data collection tools used in this study: Please clarify this by setting a paragraph for the same and make it available as a supplementary document

b. Expert opinion with key informants? questionnaires, registers consulted? Please clarify these by categorizing them

c. It is not clear how you allocated of labor time? Kindly clarify

d. Kindly clarify how you adjusted for inflation (i. e; consumer price index (CPI)), show the formula and explain the source of CPI and which year if applicable

e. Explain how did you do annuitization (Discount rate if applicable?) of assets if applicable

f. How did you count personnel resource use; it is not clear how you did it

g. The results are presented in a very summarized manner, kindly present a table showing the identification, measurement and valuation of resources in the results as well as data methods and data sources table

h. The researcher could consider stating the major driver for the cost difference among the different AG-RDTs types to avoid speculation in the section.

i. Kindly explain in a separate paragraph how you did the cost analysis.

j. Clarify any assumptions before analysis

DISCUSSION

I wonder what the positivity rate at the time of the study was. Just like the researcher indicates that many factors can influence the cost of SARS-CoV-2 diagnosis such as the epidemiological context. How can the study be compared to other studies (if any exist) based on the epidemiological context in the study period?

6. PLOS authors have the option to publish the peer review history of their article (what does this mean?). If published, this will include your full peer review and any attached files.

**Do you want your identity to be public for this peer review?** For information about this choice, including consent withdrawal, please see our Privacy Policy.

Reviewer #1: No

Reviewer #2: **Yes: **Jesse Gitaka

---

## [Editor Report · Decision Letter 1]

12 May 2023

Testing for SARS-CoV-2 in resource-limited settings: A cost analysis study of diagnostic tests using different Ag-RDTs and RT-PCR technologies in Mozambique

PGPH-D-22-01885R1

Dear Mrs Sitoe,

We are pleased to inform you that your manuscript 'Testing for SARS-CoV-2 in resource-limited settings: A cost analysis study of diagnostic tests using different Ag-RDTs and RT-PCR technologies in Mozambique' has been provisionally accepted for publication in PLOS Global Public Health.

Best regards,

Sara Suliman

Academic Editor